# Reducing Biofilm Infections in Burn Patients’ Wounds and Biofilms on Surfaces in Hospitals, Medical Facilities and Medical Equipment to Improve Burn Care: A Systematic Review

**DOI:** 10.3390/ijerph182413195

**Published:** 2021-12-14

**Authors:** Roger E. Thomas, Bennett Charles Thomas

**Affiliations:** 1Department of Family Medicine, Cumming School of Medicine, University of Calgary, Calgary, AB T2N 4N1, Canada; 2Independent Researcher, Calgary, AB T2M 1M1, Canada; bennettct@yahoo.com

**Keywords:** burns, biofilms, multiply drug-resistant organisms, health care associated infections, hospitals, medical facilities, detection, therapies, systematic review

## Abstract

Biofilms in burns are major problems: bacterial communities rapidly develop antibiotic resistance, and 60% of burn mortality is attributed to biofilms. Key pathogens are *Pseudomonas aeruginosa*, methicillin-resistant *Staphylococcus aureus*, and multidrug-resistant *Acinetobacter baumanii.* Purpose: identify current and novel interventions to reduce biofilms on patients’ burns and hospital surfaces and equipment. Medline and Embase were searched without date or language limits, and 31 possible interventions were prioritised: phages, nano-silver, AgSD-NLs@Cur, Acticoat and Mepilex silver, acetic acid, graphene-metal combinations, CuCo_2_SO_4_ nanoparticles, Chlorhexidene acetate nanoemulsion, a hydrogel with moxifloxacin, carbomer, Chitosan and Boswellia, LED light therapy with nano-emodin or antimicrobial blue light + Carvacrol to release reactive oxygen species, mannosidase + trypsin, NCK-10 (a napthalene compound with a decyl chain), antimicrobial peptide PV3 (includes two snake venoms), and polypeptides P03 and PL2. Most interventions aimed to penetrate cell membranes and reported significant reductions in biofilms in cfu/mL or biofilm mass or antibiotic minimal inhibitory concentrations or bacterial expression of virulence or quorum sensing genes. Scanning electron microscopy identified important changes in bacterial surfaces. Patients with biofilms need isolating and treating before full admission to hospital. Cleaning and disinfecting needs to include identifying biofilms on keyboards, tablets, cell phones, medical equipment (especially endoscopes), sinks, drains, and kitchens.

## 1. Introduction

Biofilms affect > 80% of bacterial infections in humans [1,2] and are the dominant mode of bacterial growth in which millions of bacteria cohabit in a hydrated extracellular matrix [1]. Biofilms are a key problem in burns, and 60% of the mortality from burns is attributed to biofilms [1]. Biofilms are also a key problem in chronic wounds such as diabetic, pressure and venous leg ulcers, lung infections in cystic fibrosis, pneumonia in patients on ventilators, and patients on medical devices and urinary catheters [1]. Most hospital-acquired infections are due to vancomycin-resistant enterococcus (VRE), methicillin-resistant *Staphylococcus aureus*, *Klebsiella pneumoniae*, multidrug-resistant *Acinetobacter baumanii*, *Pseudomonas aeruginosa*, and extended spectrum beta-lactamase-producing organisms (ESBL) [3].

Biofilms have five life stages: reversible attachment to surfaces, irreversible attachment, maturation-1, maturation-2, and the planktonic form. During the final stage, ~80% of the biomass may convert back to the planktonic form and bacterial susceptibility to antibiotics becomes similar to that of other planktonic cells [1]. Factors related to biomass dispersion are decreased availability of carbon, iron, oxygen, pyruvate and nitric oxide, oxidative stress, and starvation and decreased quorum-sensing molecules such as farnesol, indole and N-acylhomoserine lactone, and increases in bile salts and cis-2-decenomic acid [1]. 

Bacteria in biofilms can develop resistance 100- to 1000-fold against multiple antimicrobials compared to planktonic cells [4]. Gram-negative bacteria are a key problem in biofilms because they have an outer and an inner cell membrane. Penicillin class antibiotics are degraded by enzymes within bacterial cells [5], and the polar lipopolysaccharides in the outer membrane are hydrophobic to antibiotics such as rifampicin. In the membranes, impaired function of influx pumps and increased function of efflux pumps enable Gram-negatives to acquire resistance against multiple antibiotics because of the resulting sub-minimal inhibitory concentration (MIC) of antibiotics within cells. One group of efflux pumps (resistance-modulation class) effluxes antibiotics using energy from the proton motive force generated by differences in potential between ions and protons across membrane walls and interventions in cell membranes to alter these potential differences could be therapeutic [5]. Changes in membrane porins and decreased passive diffusion of antibiotics also decrease antibiotic concentrations. 

Increased resistance to antibiotics is also caused by transfers of resistance genes between bacteria, slow bacterial growth rates within biofilms, persister cells, which are particularly resistant to destruction, and quorum-sensing gene activity between bacteria [4]. Bacteria within biofilms minimise the host immune system activities by using quorum sensing genes to communicate. In *P. aeruginosa*, the *LasI* gene affects biofilm formation through the quorum sensing system and also regulates the virulence factors alkaline phosphatase, elastase, exotoxin A, pyocyanin, and rhamnolipid [2]. The *RhlR* gene uses butyryl acyl homoserine lactone to regulate its own gene and the quorum-sensing operon and the genes coding for pyocyanin, siderophores, and rhamnolipid synthesis enzymes [6]. In *A. baumanii*, the *abal* gene affects the quorum sensing system and biofilm formation through acyl homoserine lactose. In *S. aureus*, the Agr accessory gene regulator affects the quorum sensing mechanism and biofilm formation [2]. 

Burns are rapidly colonised by Gram-positive bacteria, principally *S. aureus*, from the patients’ skin and infected environmental surfaces that patients contact and then within hours to a few days the wound is colonised by Gram-negative bacteria, principally *P. aeruginosa* and *A. baumanii.* A study of burns in ICUs and burn wards identified 1621 pathogens in 2395 clinical samples, of which 74.2% were Gram-negative and 34.3% were *A. baumanii* [7]. The early treatment of burns is important to prevent colonisation by multiple bacteria, particularly *P. aeruginosa* and *A. baumanii*.

Thus the initial step in burn therapy is wound cleaning and debridement to remove necrotic tissue, which significantly reduces blood flow and immune system access. Topical dressings and antibiotics and then IV antibiotics are applied if the patient does not respond but high resistance levels to multiple antibiotics soon develop. The consequences for patients are failure of skin grafts [8], bacteraemia, infections of multiple organs, and mortality. Multiple therapies have been developed to be applied directly to burns to inhibit bacteria in both the planktonic phase and prevent biofilm formation by Gram-positives and Gram-negatives including interventions to damage bacterial cell walls, especially Gram-negative inner and outer membranes, and facilitate higher entry levels of antibiotics. More than 1000 wound models to test these therapies have been reported with 74% in vivo, 23% in vitro, and 3% ex vivo (ex vivo uses samples from living animals but conducts experiments in laboratory equipment). Pigs are the preferred animal as their skin and immune system are closer to humans than rat or mouse skin [9]. 

A systematic review of interventions in hospitals and long-term care facilities identified 14 cluster-randomised controlled trials of cleaning and disinfecting strategies to reduce the incidence of healthcare-associated infections (HAIs) and multiply drug-resistant organisms (MDROs) of particular concern: methicillin-resistant *Staphylococcus aureus* (MRSA), vancomycin-resistant enterococcus (VRE), multidrug-resistant *Acinetobacter* species, and extended spectrum beta-lactamase producing organisms (ESBL). Of fourteen c-RCTs, ten were focused to reduce patient infections by four MDROs and/or HAIs. In four c-RCTs, patient MDRO and/or HAI rates were significantly reduced with cleaning and disinfection strategies, including bleach, quaternary ammonium detergents, ultraviolet irradiation, hydrogen peroxide vapour, and copper-treated surfaces or fabrics, but in six there were no significant changes. Three c-RCTs focused on reducing MRSA rates (one had significant results), and one on *Clostridioides difficile* had no significant results. No study assessed the contributions of biofilms [10]. The disappointing results of these c-RCTs of disinfection suggest that a key preventive strategy is to first detect patients with MDROs or HAIs, isolate them in an isolation unit, and treat them before they enter other areas of medical facilities to transmit infection to patients, staff, and surfaces. Another key strategy is identifying and destroying biofilms in patients’ burns and on surfaces and medical equipment within medical facilities before patients and staff can contact them. 

Patients with burn wounds quickly acquire a range of pathogens from their own skin and their environment and if they are hospitalised for care they are very likely to acquire MDROs and HAIs from surfaces, patients, staff, and equipment in the hospital environment. Burn patients have high rates of biofilm infections, and current research on interventions to reduce biofilms in burn patients includes silver and other metals, disinfectants, hydrogels, light and sound therapy to activate sensitiser molecules to release active oxygen species, a variety of small molecules to enable better penetration of cell wall membranes, glycans, lactobacilli, and phage therapy.

A systematic review and meta-analysis of tests of antimicrobial efficacy against biofilms identified five key model parameters that influence outcomes: the biofilm surface area/volume ratio, biofilm areal cell density, fluid static or flow conditions over the biofilm, biofilm age, and the antibiotic chosen as the comparator. The review compared several pairs of studies and concluded that the experimental method used is the most important factor determining the outcome and presented comparisons of studies that showed that the method chosen can “produce extremely different results even for the same microbial agent.” [11]. 

A systematic review of topical agents for managing chronic biofilm infections identified 39 in vitro, 5 animal and 3 human in vivo studies involving 44 commercially available topical agents and 78 biofilm-forming bacteria and concluded: 

“The analysis clearly identifies a large disparity in the translation of laboratory studies to researchers undertaking human trials.” “When analysing the thirty-nine included in vitro studies, a standardised methodological approach to biofilm testing was not observed. Sixteen different biofilm models were used with significant variations between test parameters such as: choices of different bacterial strain or isolate (n = 78), biofilm growth time (24 h to 168 h), starting log densities, agent exposure duration (3 s to 168 h), and adaptation to in vitro models to more closely resemble a wound environment (n = 22 of 39, 56%).” [12], pp. 266–267).

Purpose: To assess if any current or under development therapies intended to be used directly on patient burns have been tested in well-designed trials and have significant outcomes so that they merit large c-RCTs to test benefits for patient outcomes and also current cleaning and disinfection strategies of surfaces and medical equipment in hospitals and medical facilities to improve outcomes for burn patients.

## 2. Materials and Methods

Literature searches were conducted on 1 December 2021 in Medline, Embase, Cochrane Central, and Web of Science from inception with no language or date limits using the search terms (biofilm) and (hospital or long-term care facility or medical facility) and (burns). Separate searches were undertaken for (1) (medical equipment or exp stethoscopes or keyboard.mp or exp computer or exp telephone or exp Cell Phones or computer tablets.mp or exp computers Handheld or endoscopes) and (biofilm) and (detection or measurement); and (2) (burns) or (hospitals or long-term care facilities or medical facilities) and (systematic reviews or meta-analyses). Titles and abstracts were assessed and data abstracted independently by two researchers with disagreements resolved by discussion. The systematic review is registered with Prospero and follows the PRISMA reporting requirements 

## 3. Results

### 3.1. Literature Search

The literature search identified 31 studies of novel interventions to prevent biofilm formation (Figure 1).

The studies chosen for analysis are those with the most direct relevance to burn care: in vivo human biofilm studies then in vivo studies of pigs (their skin and immune system are closer to human skin than other animals) and then in vivo studies of rats and mice. In vitro studies of pathogens from burn wounds grown in 96-well microplates do not provide as direct evidence because they are not in living mammals and are on abiotic surfaces. All the included studies were from Medline and Embase. 

### 3.2. Interventions

#### 3.2.1. Silver

Silver sulfadiazine has been the principal burn topical therapy for decades. Silver in burns absorbs wound exudates and kills organisms drawn into the dressings; binds to negatively charged proteins, RNA and DNA; damages bacterial cells walls; inhibits replication; and reduces metabolism and growth [13]. With the development of antibiotic resistance including plasmid-mediated resistance, new therapies have been developed which enhance the entry of silver into burn wounds. Gholamrezazadeh found that a nanomolecule formulation of silver (nano-Ag) at 12.5 ng/mL caused a reduction in the number of *P. aeruginosa* bacteria forming biofilms from 28.5% to 3.5%, whereas benzalkonium MIC at 0.03 mg/mL reduced the number of bacteria forming biofilms from 28.5% to 18.7% [6].

Pourhajibagher sensitised liposomes with the photosensitising agent Cucurmin to create the reactive oxygen species (ROS) superoxide anion O*_2_¯, hydroxyl radical *OH and singlet oxygen ^1^0_2_ which decrease the expression of bacterial virulence genes. Cucurmin also enables repair and regeneration of damaged eukaryotic cells. In in vitro biofilms with MIC_90_ doses of silver sulfadiazine nanoliposomes with Curcumin (AgSD-NLs@Cur) activated by light diodes, the cell numbers of *A. baumanii* in biofilms decreased by 76.4%, with silver sulfadiazine nanoliposomes (AgSD-NKs) by 44.8%, and with silver sulfadiazine (AgSD) by 38.1%. In vivo mouse burn wounds treated with AgSD-NLs@Cur on staining and with light microscopy showed focal epidermis regeneration, fibrosis, and granulation tissue formation, but the controls showed complete loss of the epidermis and hair follicles, hyperaemic vessels, and extensive bacterial colonisation [14]. 

A comparison of eleven antimicrobial burn dressings found that 72 h after infection with *A. baumanii*, two silver formulations reduced *A. baumanii* numbers in biofilms (commercial silver formulations Acticoat by 96%, Mepilex Ag by 95.9%) and acetic acid by 90–93%. *P. aeruginosa* numbers were reduced by Acticoat by 100%, Mepilex Ag by 100% and acetic acid by 86–93% [15]. 

#### 3.2.2. Other Metals

Karaky’s in vitro study found that eight metal-graphene combinations reduced *P. aeruginosa* biofilms by ≥90% (platinum-graphene oxide, gallium-graphene oxide, molybdenum-graphene oxide, gold-graphene oxide, silver-graphene, gallium-graphene, and molybdenum-graphene). The greatest reduction of biofilm metabolic activity occurred with gold-graphene oxide (94%), molybdenum-graphene oxide (93%), silver (91%), and silver-graphene (91%) [8]. Li assessed the effect of H_2_O_2 (_2 nM) and CuCo_2_S_4_ nanoparticles (100 μg/mL) on mouse burns, and after two days of treatment there was no inflammatory response. The burn wounds contracted whereas the control group showed a severe inflammatory response with suppuration. After six days the group treated with H_2_O_2_ at 2 nM and CuCo_2_S_4_ at 100 μg/mL showed enhanced healing and 83.7% wound closure, with CuCo_2_S_4_ 71%, with H_2_O_2_ 63.3%, and the control 59%. At two weeks, the H_2_O_2_, and CuCo_2_S_4_- treated wounds were completely closed and healed [15]. 

Nozari compared chitosan/alginate + ZnO nanoparticles to chitosan/bentonite + ZnO nanoparticles and noted ~1 × 10^4^ lower cfu/mL rates of *S. aureus* and *P. aeruginosa* compared to control (99.99% reductions). With in vivo mouse burns at 7 days for the treated rats there were re-epithelialisation, active fibroblasts, and hair follicles, and sebaceous glands were detected but there was no re-epithelialisation in untreated rats [16].

#### 3.2.3. Disinfectants

Halstead assessed the effect of acetic acid at concentrations ranging from 0.31% to 5% against 23 isolates of 6 MDROs and in vivo biofilms; the acetic acid MBIC was 0.31%, the MBEC against formed biofilms ranged from ≤0.10% to 2.5% and eradication of mature biofilms was observed for all isolates after three hours of exposure [17]. 

Song treated mouse burn wounds with Chlorhexidene acetate nanoemulsion (CNE) 2 μg/mL, and according to scanning electron microscopy MRSA biofilms were “dispersed and disrupted and obvious reduction in number of bacteria,” there were large vacuoles between the cell wall and cytoplasm. There was also leakage of DNA, proteins, K^+^, and Mg^2+^. The dead/live cell ratio with CNE was 83.6% [18].

Tiwari compared the effects of sodium hypochlorite and ethanol on *S. aureus* in vitro biofilms and found no significant differences in reductions of strong compared to weak *S. aureus* biofilm formers. However, electron microscopy of strong biofilm producers showed significant depressions and irregular craters on their surface [19]. 

#### 3.2.4. Hydrogels

Andersson’s study of Göttingen minipig burn biofilms compared Prontosan (0.1% polyhexamethylene biguanide and 0.1% undecylenamidopropyl betaine) or levofloxacin on *S. aureus* and *P. aeruginosa* and on scanning electron microscopy both antibacterial treatments “visibly reduced” the number of bacterial cells on the wound surfaces and that perturbations and bacterial clumping and debris were noted on the Prontosan-treated biofilms. The bioluminescence levels of luminescent *S. aureus* and *P. aeruginosa* at two hours after antibiotic therapy were significantly reduced by Protosan (*p* < 0.001 to 0.0001) [9]. 

Chhibber treated MRSA burn wounds with a hydrogel (moxifloxacin 0.5% *w/v*, carbomer 1% *w/v*, Chitosan 5 mg/mL, Boswellia gum 0.5%) and showed a 3.5 log_10_ reduction on day 1 and a 4.8 log_10_ reduction on day 2, whereas the control increased to 6.9 log_10_ on day 3. At 4 h, there was complete eradication of MRSA from the wounds but MRSA was established in the control mice. At 24 h there was subdued inflammation and signs of healing in the treated mice but in the control mice there was loss of epithelium, proliferation of neutrophils, and a thick layer of inflammatory cells [20]. 

#### 3.2.5. Sound

Pourhajibagher used ultrasound on nanoemodin (2.5 × 10^−4^ g/L) to release oxygen species (ROS) including superoxide anion O*_2_¯, hydroxyl radical *OH, and singlet oxygen ^1^O_2_, which damaged bacterial cell membranes, proteins, and DNA. There was a reduction in *Staphylococcus aureus*, *Pseudomonas aeruginosa*, and *Acinetobacter baumannii* following sonotherapy at 1/2 MIC of N-EMO of 81.5%; at 1/16 MBIC 71.0%; and at 1/128 MBEC 57.8 with reductions in log_10_ cfu/mL of 99.99%, 99.97%, and 99.48% but no effect with ultrasound alone [21]. 

#### 3.2.6. Light

Ishiwata compared ethylene diamine-tetra-acetic acid disodium salt (EDTA), which suppresses biofilms, and dimethyl sulfoxide (DMSO), which is a tissue-penetration enhancer on rat burns. Only 2/14 of the rats survived to 7 days, but 11/14 of those also exposed to methylene blue 665 nm LED diodes at 45 mW/cm^2^ at 2.5 cms three times daily for 20 min × 7 days, presumably because the recurrent reductions in the *P. aeruginosa* levels gave the LED-treated rats a better chance [22]. 

Lu used the phytochemical carvacrol at 0.2 mg/mL and blue light 450 nm at 75 J/cm^2^ to excite porphyrin-like derivatives in bacterial cells to produce reactive oxygen species (ROS). In vitro biofilms of *Acinetobacter baumanni*, *Pseudomonas aeruginosa*, and MRSA at 10^7^ CFU were completely eliminated (*p* < 0.0001). The thickness of *Acinetobacter baumanni* biofilm was reduced from 58.6 μm to 1.4 μm and MRSA biofilm from 32.4 μm to 1.7 μm. In in vivo mouse burns infected with *Acinetobacter baumainii* at 5 × 10^5^ cfu/mL, carvacrol 50 μL at 1 mg/mL and blue light for 12 min (40 J/cm^2^) eliminated log 8 of luminescence in luminescent bacteria, blue light alone 2.3 log, and carvacrol 0.8 log. Reactive oxygen species increased 14-fold in the *Acinetobacter baumanni* group, 12-fold in the *Pseudomonas aeruginosa* group, and 8-fold in the MRSA group [3]. 

Pourhajibagher exposed *Acinetobacter baumannii*, *Pseudomonas aeruginosa*, and *Staphylococcus aureus* to photodynamic therapy (aPDT) with the photosensitiser indocyanine green (ICG) at 1000 μg/mL and a diode laser at 810 nm. There was a significant reduction in cell viability of *A. baumannii* to 1.5 × 10^5^ cfu/mL, *P. aeruginosa* to 1 × 10^5^ cfu/mL and *S. aureus* to 1.0 × 10^5^ cfu/mL compared to control at 4.5 × 10^5^ cfu/mL (all *p* < 0.05). There was also a 54% increase in reactive oxygen species (ROS) compared to controls and decreases in the expression of the *P. aeruginosa* quorum-sensing *abal* gene by 1.9-fold, *agrA* by 3.7-fold and *lasI* by 4.9-fold. As shown by scanning electron microscopy with a diode laser + ICG there was a reduction in cell size and numbers, cell elongation, and increased cell destruction but no change with the diode laser or ICG individually [2]. 

Wang exposed 72 h old *A. baumanii* biofilms to antimicrobial blue light (aBL) 432 J/cm^2^ for 72 min, which resulted in the inactivation of 3.18 log_10_ cfu/mL and the exposure of *P. aeruginosa* biofilms to aBL in the inactivation of 3.12 log_10_ cfu/mL for 72 h old biofilms, but control biofilms showed only a <0.27 log_10_ cfu/mL loss of viability for *A. baumanii* and <0.42 log^10^ cfu/mL for *P. aeruginosa.* An in vivo study of mouse burn wound biofilms infected with 5 × 10^6^ cfu/mL *A. baumanii* at 24 h required 360 J/cm^2^ at 48 h 540 J/cm^2^ to inactivate 3 log_10_ cfu/mL [4].

### 3.3. Small Molecules

**Mannosidase and trypsin enzymes** Mannosidase and trypsin enzymes attack the biofilm matrix of *P. aeruginosa*, which consists of three layers: alginate (a polymer of β-d-mannuronic acid and α-l-glucuronic acid which provides structural stability and protection of the biofilm); a repeating polysaccharide of D-mannose, D-glucose, and L rhamnose with an important role in biofilm formation and protection; and a glucose-rich layer (Pel). Banar compared ceftazidime (CAZ) 1024 μg/mL, CAZ + α-mannosidase 4 μg/mL, CAZ + β-mannosidase 4–8 μg/mL, and CAZ + trypsin 8–32 μg/mL, and all combinations killed bacterial biofilm cells at these concentrations [23]. 

**Antimicrobial compounds with an aromatic naphalene (N) or benzene (B) core, a L-lysine moiety and a variable lipophilic chain** Ghosh found that a compound with a naphthalene core and a decyl chain appendage (NCK-10) was the most active against NDM-1-producing Gram-negative pathogens. For in vitro biofilms of *A. baumanii*, *E. coli*, *K. pneumoniae*, and *P. aeruginosa*, the MIC for NCK-10 was 4.5 μg/mL. NCK-10 completely lysed persister cells of 5 log cfu/mL *E. coli* after 2 h but colonies persisted in the control group at 5 log cfu/mL. To disrupt biofilms, the EC_50_ was 30 μM against biofilms of *A. baumanii* (MTCC 1425), 20 μM against *E. coli* (MTCC 443), *26* μM against *K. pneumoniae* (ATCC 700603), and 19 μM against *P. aeruginosa* (MTCC 424). On confocal microscopy in the treated samples the biofilms were completely disrupted. NTK-10 did not induce bacterial resistance (there was no change in MIC after 20 passages) but the MIC of colistin increased 250-fold. In burn wounds of mice there was significant reduction in bacterial burden after daily topical treatments with 40 mg/kg × 7 days compared to control [24].

**Pyruvate-dehydrogenase** PDH catalyses pyruvate to acetyl-CoA in the presence of CoA and NAD^+^ and the microcolony formation factor MifR. Goodwine assessed if enzyme pyruvate-dehydrogenase (PDH) would increase the efficacy of tobramycin killing of biofilms of *P. aeruginosa* and *S. aureus* strains from wound debridement samples. In in vitro human wound samples, there was a 2.2-fold reduction in bacteria after exposure to 5 mU DPH and 2.9-fold reduction after 10–20 mU. In in vitro biofilms investigated with confocal laser scanning microscopy, 60% of microcolonies in PDH-treated biofilms showed signs of dispersion with central voids, but only 8% of untreated biofilms [1].

**Li-F type peptide AMP-jsa9** Han assessed whether the Li-F type peptide AMP-jsa9 (which both kills planktonic cells and penetrates MRSA cell membranes) would reduce *S. aureus* biofilms. Cell viability was reduced to 10% with 8 × MIC vancomycin (8 μg/mL) and to 10% with AMP-jsa9 at 8 × MIC (128 μg/mL). Biomass was reduced to 15% with 8 × MIC Vancomycin (8 μg/mL) and to 15% with AMP-jsa9 at 8 × MIC (128 μg/mL). The viable cell counts in mouse skin burns treated with vancomycin or AMP-jsa9 were 10^1^ to 10^2^ on days 3 and 7 and at 3 days compared to those treated with kanamycin 2–3 × 10^4^ or saline 5–6 × 10^5^ [25].

**D-LANA-14**. D-lysine conjugated aliphatic norspermidine analogue with a tetradecanoyl chain, which can depolarise Gram-negative cell membranes. Konai found D-LANA-14 was “moderately active” at MICs 32–64 μg/mL against three strains of *A. baumanii* and four strains of *P. aeruginosa*, but the combination of D-LANA-14 at sub-MIC levels enabled tetracycline at 4 μg/mL and rifampicin at 2 μg/mL to be active against both bacteria. Against in vitro biofilms, confocal scanning electron microscopy showed that D-LANA-14 (8 μg/mL) plus colistin (8 μg/mL) resulted in >80% reduction in the biofilm mass of *A. baumanii*-R674 and *P. aeruginosa*-R590, but D-LANA-14 (8 μg/mL) alone showed no effect, and rifampicin (8 μg/mL) 25–30% disruption. In burn wounds in mice D-LANA-14 (40 mg/kg) plus rifampicin (40 mg/kg) caused a 4.9 log reduction in *A. baumanii*-R674 and a 4.0 log reduction in *P. aeruginosa*-R5902, but for D-LANA-14 alone it was 2.3 log and 1.3 log, and for rifampicin alone it was 3.0 log and 1.6 log [5].

**PV3 antimicrobial peptides** can disrupt negatively charged bacterial cell membranes. PV3 includes terminal residues from two snakes: pEM-2 from *Bothrops asper* and Mastopparan-VT-1 from *Vespa tropica.* For multi-drug-resistant strains of *P. aeruginosa* from burn wounds of hospitalised patients in Iran, the MIC and MBC of PV3 were 2–4 μg/mL, and for ceftazidime, the MIC was 16–256 μg/mL. For the in vitro biofilms, PV3 at 8 × MIC at 24 h resulted in “almost” 100% killing of cells and 95% biomass removal and scanning electron microscopy of the results showed that the PV3-treated cells were shorter, with blisters on membranes, roughness, and blebbing [26]. 

**Poly(l-ornithine)s and poly(l-lysines)** Pan assessed the ability of amino-acid-based star-shaped poly(l-ornithine)s and poly(l-lysines) with varying surface charge/hydrophobicity balances (P03, PL2, PH3) to disrupt bacterial cells and biofilms. In vitro P03 reduced the biomass of *P. aeruginosa* biofilms by 76.9%, PL2 by 35.1%, PH2 by 31.45%, and Polymixin by 7.8%. In mouse burn wounds, P03 caused a 78.2% reduction in *P. aeruginosa* and PL2 caused a 49.3% reduction compared to Polymixin B [27]. 

**Platensimycin** (PTM) and Platensimycin-thioether analogues (PTM-**2t**) target the Fab/FabF of bacterial fatty acid synthases. For in vitro biofilms of *S. aureus*, ATCC 291213 after treatment with 2 μg/mL of PTM or PTM-**2t** the biofilm was reduced by 95%. For the mouse burns treated with 4 mg of PTM or PTM-**2t** cream on the wound twice daily × 7 days, PTM reduced *S. aureus* to 2 × 10^6^ cfu/g and PTM-**2t** to 8.6 × 10^6^ cfu/g compared to 2.5 × 10^6^ cfu/g for mupirocin and untreated mice 4.3 × 10^8^ cfu/g. On haematoxylin and eosin staining, untreated mice showed partially destroyed hair follicles, an incomplete fat layer, and a large number of inflammatory cells in the muscle layer, but treated mice had “relatively healed skin structure” [28].

**Salicylidene acylhydrazide** INP0341 was assessed for its ability to inhibit the *P. aeruginosa* type III secretion system (T3SS) which translocates virulence factors; the four exoenxyme (Exo) molecules S, T, U, and Y from the bacterial cytosol directly into the host cytoplasm where they alter cell function to permit bacterial growth; and the flagellum system required for biofilm formation and motility. In *P. aeruginosa*, flagella are required for swimming, and flagellate and type IV pili for swarming. In vitro INP0341 significantly reduced in a dose-dependent manner the expression and secretion of the Type III secretion system T3SS ExoS required by *P. aeruginosa* for colonisation and survival in host cells. INP0341 disarmed but did not kill *P. aeruginosa* cells. In in vitro biofilms, INP0321 at 100 μM reduced biofilm mass to 40% of control (*p* < 0.05), inhibited *P. aeruginosa* swarming, and prevented movement across semisolid surfaces, which requires flagella and type IV pili. In vivo, the treated mice died at 36 h and controls at 42 h (*p* < 0.05) [29].

### 3.4. Glycans

Mucus lines all wet epithelial cells in the human body, including lungs, gastrointestinal and urogenital tracts, and eyes. It is the body’s first line of defence against pathogens and is occupied by trillions of sensing bacteria and white cells. Mucins suppress virulence genes, encourage the planktonic state in bacteria, prevent attachment to surfaces, and reduce bacterial toxicity to human cells. Mucins have many complex glycan structures covalently linked to serine and threonine and have been assessed as potential methods of decreasing biofilm formation. Wheeler exposed *P. aeruginosa* PA01 biofilms to MUC5AC glycans from fresh pig intestines, MUC2 from pig stomachs, and MUCB human salivary glycans. *P.*
*aeruginosa* PA01 biofilms were exposed to these mucins, and 70% of the cells dissociated from surfaces into the planktonic phase (*p* < 0.0001). Intestinal mucins suppressed quorum sensing (*lasR*), siderophore biosynthesis (*pvdA*), and type-three secretion (*pcrV*) genes. MUC5AC and MUC5B (0.5% *w*/*v*) suppressed virulence pathways 1, 2, 3, and 6 secretion systems; siderophore biosynthesis; pyoverdine and pyochelin; and quorum sensing. MUC5AC suppressed *P. aeruginosa* PA01 attachment to plastic and glass surfaces and attachment to live HT human epithelial cells in a concentration-dependent manner. In pig burn wounds injected with *P. aeruginosa* and PA01 (1 × 10^5^ cfu/mL) and treated with MUC5AC one week later, there were two-log reductions in *P. aeruginosa* cfu/mL but no reduction without mucins [30].

### 3.5. Lactobacilli 

Lactobacilli enhance phagocytic activity, inhibit neutrophil and macrophage apoptosis, produce lactic acid (which reversibly damages DNA, RNA, and proteins within *E. Coli*, *P. aeruginosa and S. enterica*) and also produce hydrogen peroxide, which reduces pyocyanin, elastase, and rhamnolipid produced by *P. aeruginosa. Lactobaccillus gasseri* supernatant inhibited the growth of *P. aeruginosa* strain PAO1 on mouse burn wounds, reduced biofilm development 40-fold at 8 h (the control increased significantly), and eliminated biofilms at 28 h. Treatment of the wound did not inhibit *P. aeruginosa* growth at 24 h but prevented transfer to the blood stream, liver, or spleen and 100% of the mice survived at 7 days. A second dose of the supernatant 24 h after the first dose completely eliminated *P. aeruginosa* in the wounds. In untreated mice, the death rate due to sepsis was 100%, and the mice had ~10^7^ cfu/mL *P. aeruginosa* g^−1^ in their livers and spleens [31].

### 3.6. Bacteriophages 

A systematic review of 95 studies of phage therapy concluded that phage therapy provided 100% protection against MDRO infections and that bio-sanitisation of foods, beverages and surfaces was 100% successful [32]. Alves injected partial-thickness second-degree burn wounds on pig skin (not live pigs) with MRSA252-Rif (resistant to rifampicin) at 10^4^ cfu/mL and then exposed the wounds to Phages DRA88 and SAB4238-A at 10^9^ pfu/mL. On the ex vivo biofilms 24 h after phage treatment, the phage-treated pigskins had 10^6.5^ MRSA252-Rif cfu/mL compared to the control at 10^7.5^ cfu/mL (*p* ≤ 0.0001). At 48 h after phage treatment, the phage treated pigskins had 10^7^ cfu/mL MRSA252-Rif compared to control 10^7^ cfu/mL (n.s.). Although the results were non-significant at 24 h, there was a 17.25-fold increase in phage numbers and at 48 h a 64.6-fold increase. Why phage numbers increased but infections were not controlled merits investigation [33].

Ho, in four ICU rooms in a 6-month intervention in a 945-bed Taiwanese teaching hospital, used aerosols of eight phages (5.5 × 10^4^ pfu/cm^2^) and found that Carbapenem-resistant *Acinetobacter baumanii* (CRAB) rates declined from 8.57/1000 patient-days pre-intervention to 5.11 during the aerosol intervention period (*p* = 0.0029). Resistant isolates decreased from 87.6% to 46.07% (*p* = 0.001). Colistin decreased from 7876 DDD/1000 patient-days to 3158 (*p* =0.0177), tigecycline 2737 to 753 (*p* = 0.0005), meropenem 5084 to 2469 (*p* = 0.0385), and imipenem 1384 to 1101 (ns) [34].

Holguín exposed three multiply-drug-resistant *P. aeruginosa* strains to the Φ*Pan70* phage at 6.5 × 10^7^ pfu/mL. Eighteen hours after phage therapy, *P. aeruginosa* P1 decreased from 10^7.5^ to 10^4^ cfu/mL, P2 10^8^ to 10^4.5^, and P4 10^7.5^ to 10^2.5^. In the in vitro biofilms for P1, there was a 17% reduction at 0 h (*p* = 0.003), 34% at 24 h (*p* = 0.134), and 55% at 48 h (*p* = 0.005); for P3 a 59% reduction at 0 h (*p* = 0.00001), 56% at 24 h (*p* = 0.034), and 75% at 48 h (*p* = 0.0004); and for P4 a 68% reduction at 0 h (*p* = 0.015), 15% at 24 h (*p* = 0.036) and 21% at 48 h (*p* = 0.286). When Φ*Pan70* was given to the mice immediately after *P. aeruginosa* infection, 4/5 mice survived, and for those who received Φ*Pan70* 45 min after infection, 5/5 survived. For those who received phage therapy 24 and 48 h after infection, 4/5 mice survived, but in the control, all mice died on days 3 or 4 [35].

O’Flaherty in an in vitro study of 28 *S. aureus* strains from outpatients, inpatients, and hospital staff in Ireland over a 3-year period, found that 14/28 of these *S. aureus* strains were sensitive to phage K at 10^7^ cfu/mL, no bacteria remained after 2 h, and there were no bacteriophage-insensitive mutants after 25 h. MRSA strain DPC5645 was reduced within 2 h from 5.7 × 10^6^ cfu/mL to undetectable levels, and in vivo MRSA strain DPC5246 on skin was reduced 100-fold with phage K 1.4 × 10^8^ pfu/mL [36].

Pallavali for *P. aeruginosa*, *S. aureus*, *K. pneumoniae*, and *E. coli* used bacteriophages at 1 × 10^9^ pfu/mL for in vitro biofilms and after 4 h of phage therapy at 96 h, the optical density (which corresponds to biomass) for *P. aeruginosa* was 0.47 ± 0.035 and decreased to 0.17 ± 0.024; for *E. coli* it was 0.47 ± 0.035 and decreased to 0.15 ± 0.026, for *K. pneumoniae* it was 0.47 ± 0.035 and decreased to 0.17 ± 0.022, and for *S. aureus* it was 0.47 ± 0.036 and decreased to 0.16 ± 0.032. In vitro confocal microscopy showed “predominant numbers of dead cells” after 4 h of phage therapy [37].

#### 3.6.1. Risk of Bias Assessment: Numbers of Bacterial Strains Tested, Numbers of In Vivo Tests Using Animals, and Summary Measures Used in In Vitro and In Vivo Biofilm Outcomes

Authors often tested few bacterial strains, many made no animal studies, and in the animal studies numbers were very small. Of the 31 studies, 6 tested 1 strain of *Acinetobacter baumanii*, 1 tested 2 strains, 2 tested 3 strains, 1 tested 8 strains, and 1 tested 100 clinical isolates from burn patients. Twelve studies tested one strain of *Pseudomonas aeruginosa*, two tested two strains, one tested three strains, one tested seven, one tested nine, one tested twelve, one tested twenty-eight strains, and one tested fifty-seven clinical isolates from burn patients. Many tested one strain (PA01), which is not usually pathogenic in humans without risk factors. Eight studies tested one strain of MRSA, one tested twenty-seven, and one tested thirty-six strains. One study tested one strain of *Klebsiella pneumoniae*, and two tested two strains. Four strains of *E. coli* were tested. Three studies tested burns on 20 mice, one on 32, one on 35, one on 54, and seven studies did not state the number of mice. One study tested 3 rats and another tested 4 rats. One study tested 3 pigs, one 4, one did not state the number, and one used pigskins and not live pigs (Table 1). 

The multiple outcomes used by authors made summaries difficult and no meta-analysis could be undertaken. In particular, the variety of microscopy techniques to measure changes in biofilms could not be summarised. Fourteen studies used one or more of these as their main outcome measure: % reductions in the numbers of bacteria forming biofilms or cell counts, intact biofilms, bacterial viability, or biofilm optical density. Of these, 13 reported cfu/mL; 11 reported MICs; 5 reported Minimal Biofilm Inhibitory Concentrations (MBICs); and 3 reported fold-reductions in gene expression or production of reactive oxygen species (ROS). Some reported electron scanning and other microscopy techniques of the effects of interventions on biofilms. Two studies reported if the bacteria were strong, intermediate, or weak biofilm formers [19,23], but in other studies, if there were weak bioformers an intervention could be incorrectly assessed as being the cause of biofilm reduction. The reporting of the technical details of the preparation of samples and measurement methods was extensive, whereas few tested interventions at different strengths or replicated their assays (Appendix A).

#### 3.6.2. Identification of Candidate Interventions for Further Testing in Large Scale c-RCTs

It is not possible to identify a simple metric such as reduction in cfu/mL by which to rank studies for further large-scale testing because of the heterogeneity of bacteria tested, whether dose ranging studies were conducted, the number of replications of experiments, whether tested in animals, ability of bacteria to form strong biofilms, biofilm outcome measures, and reporting of scanning electron microscopy of biofilms. Interventions meriting further testing on the basis of one or more outcome measures have been starred *** in Table 1. These include Gholamrezazadeh’s study of nano-silver [6], Pourhajibagher’s study of AgSD-NLs@Cur [14], Halstead’s studies of silver (Acticoat, Mepilex) and acetic acid [13,17], Karaky’s study of graphene–metal combinations [8], Li’s study of CuCo_2_S_4_ nanoparticles [15], Song’s study of Chlorhexidene acetate nanoemulsion (CNE) [18], Chhibber’s study of a novel hydrogel (moxifloxacin, carbomer, Chitosan, and Boswellia) [20], Pourhajibagher’s study of nano-emodin to release reactive oxygen species (ROS) [21], Lu’s study of blue light + carvacrol [3], Wang’s study of antimicrobial blue light [4], Banar’s study of mannosidase + trypsin [23], Ghosh’s study of NCK-10, a naphthalene core compound with a decyl chain appendage [1], Han’s study of AMP-jsa9 [25], Ghosh’s study of NCK-10 [24], Konai’s study of D-LANA [5], Memariani’s study of the antimicrobial peptide PV3, which includes two snake venoms [26], Pan’s study of the polypeptides P03 and PL2 [27], and the studies of phage therapy by Ho [34] and O’Flaherty [36]. Replications in large RCTs using animal models then in c-RCTs of patients, surfaces in hospital, and medical facility and medical equipment and using uniform experimental methods would best be accomplished by a large consortium of burn researchers. 

## 4. Discussion 

Bacteria are able to colonise and form biofilms on patients, especially those with burns or other wounds, who are immunosuppressed, are frail, have multiple co-morbidities, or have indwelling devices. Bacteria are able to colonise and form biofilms on multiple surfaces in medical facilities, on surfaces that patients contact and on medical equipment. Thus, a comprehensive and integrated approach is needed to identify and test interventions to treat and eliminate biofilms in all of these areas. Burn patients are at risk because they are colonised with Gram-positives and then with multiply-drug-resistant Gram-negatives. Many interventions both current and at the laboratory stage are potentially applicable to both patients and the surfaces and equipment in hospitals they are admitted to. These interventions have been starred *** in Table 1 although the studies are small. 

More intensive research is also needed to increase the effectiveness of current cleaning and disinfecting routines by making surfaces in medical facility patient rooms, treatment and common rooms, and kitchens less hospitable to bacteria and viruses by plating them with copper, silver, titanium, and other metals; impregnating curtains, bedding, and gowns with antibacterial chemicals; using ultraviolet light to disinfect keyboards on computers, phones, cell phones, and shoes, which transfer pathogens between floors and rooms. 

If the following surfaces are not included in current cleaning and disinfection routines, they need to be added and carefully monitored: sinks, drains, and toilets with their constant fluid flows are good environments for biofilms, and aerosols result when fluids are poured into them. One study of a hospital and its related LTCF found that a peracetic acid and hydrogen peroxide foam reduced Gram-negatives for an average of three days, but repeated treatments were needed [38]. Kitchens and food service also need to be included in disinfection routines and monitoring. 

A review of infection and prevention guidelines (search to April 2019) identified 31 guidelines with 1855 recommendations and rated 28 (1.5%) of the recommendations as based on systematic reviews and meta-analyses of RCTs, with 13 recommendations for devices and 311 (16.8%) recommendations as based on well-designed RCTs with strong recommendation from high-quality evidence with 64 recommendations for devices. However, only six guidelines had a GRADE recommendation. A major initiative is needed to assess the strength of evidence for cleaning medical equipment [39]. 

Cleaning biofilms from endoscopes is a major concern. A study of *P. aeruginosa* biofilms showed that they were eliminated only with 2500 ppm of peracetic acid, whereas planktonic cells were eliminated with 20 ppm [40]. A study of a quality circle to improve biofilm removal from urological endoscopes used 500 relative light units/piece as a measure of endoscopic biofilm clearance and noted an improvement from 50% to 90% when the quality circle recommendations were followed [41]. Enzymatic and alkaline detergents, bristle brushes, and Pull Thru channel cleaners to sterilise endoscopes were compared to a water flush for 108 cfu/mL of *P. aeruginosa* and *E. faecalis*, and it was found that friction applied to all surfaces of the endoscope was crucial to remove enough biofilm, and then glutaraldehyde could kill the remaining microorganisms [42]. A study of argon-plasma-activated gas on endoscopes contaminated with biofilms of MRSA, *S. aureus*, *P. aeruginosa*, or *E. coli* found that there was >8 log reduction in viable cells and the dispersal of 24 and 48 h biofilms of all bacteria [43]. 

Pathogens are transferred between rooms on the shoes of staff and patients. After shoe soles were decontaminated with ultraviolet light, there were significant reductions (*p* < 0.01) on floors, beds, furniture, and patient dummies of log_10_ 2.8 for *E. coli*, *S. aureus*, and *E. faecalis* but not for *C. difficile* [44]. 

Beds and mattresses are the surfaces patients spend most time in contact with. They often become soiled and are a good residence for bacteria. A small study of a surgery ward randomised beds to launderable or non-launderable covers and the launderable covers had significantly lower cfu counts/30 cm^2^ both after admission and discharge (*p* < 0.001) [45]. On a 36-bed medical ward 77% of bedside surfaces in contact with MRSA-negative patients were MRSA positive and 83% of bedside surfaces of MRSA-positive patients. One hour after hypochlorite disinfection, 7.4% of the bedside tables and 17.6% of the bedrails were MRSA positive, but four hours later, bedside surface contamination increased by 80% (*p* < 0.01). Using hypochlorite plus nano-organosilicon quaternary ammonium chloride spray, the MRSA level by midday declined from 4.4 ± 8.7 cfu/cm^2^ to 0.07 ± 2.6 [46]. Two LTCFs laundered bed covers with, chlorine, detergent and hot water and the *C. difficile* rate decreased by 49% [47].

This review found no RCTs of interventions to assess if there were biofilms on computers, phones, tablets, or cell phones, which are widely used by hospital and LTCF staff. Computers are used frequently by HCWs for patient care, and physicians frequently search their tablets and cell phones for data. A study of hand calculators cleaned with QUAT found after an average of 73 keystrokes that 80% of the QUAT had been removed [48]. More effective was ultraviolet light: a study of clinicians’ smartphones and wearable devices detected pathogenic bacteria on 20%, but there was a significant reduction to 4% after 30 s of ultraviolet light (*p* = 0.002) [49]. However, none of these studies mentioned or researched biofilms, and an essential part of an integrated approach to removing biofilms is to test all of these surfaces for biofilms and then set in place monitoring processes to remove and ensure that surfaces remain free of biofilms. 

The key problem thus becomes to comprehensively and accurately detect biofilms on surfaces in medical institutions (especially high touch surfaces) and on medical equipment. The current research methods of identifying and characterising biofilms are to take samples, culture them, and submit them to PCR testing and microscopy. Staining methods include the nucleic acid dye SYT09, which penetrates the membranes of bacterial cells and attaches to the DNA of both live and dead cells and fluoresces green, and the PI dye, which attaches only to dead bacterial cells and fluoresces red [50]. A key issue is whether interventions designed to identify biofilms in human burns and other wounds can also be used cost-effectively to detect biofilms on surfaces and equipment. 

Other approaches are to blot or to sonicate surfaces and wounds. A small study blotted rats’ burn wounds with a membrane to take a sample of the biofilm infected with *Ps. aeruginosa* PA01 and found that the alcian blue stain correlated 100% with the native PAGE (polyacrylamide gel electrophoresis) test, which quantifies total biofilm biomass [51]. A study of orthopaedic screws sonicated and cultured the *S. aureus*, *P. aeruginosa*, and *C. albicans* in the biofilms and found that MALDI-TOF mass spectrometry detected microorganisms with 99.9% reliability [52]). 

The lengthy process of culturing biofilm organisms may be bypassed with optical methods. A review assessed multiple methods of optical identification of bacteria including infrared spectroscopy, Fourier-transformed infrared spectroscopy, ultraviolet resonance, Raman spectroscopy, surface-enhanced Raman spectroscopy, fluorescence spectroscopy, and optical coherence tomography and concluded that Fourier-transformed infrared spectroscopy provides superior use of data particularly linked to large databases of bacterial characteristics [53]. 

Bacteriological typing methods include cultures, amplification, fragment, genome, and sequence methods. Amplification methods include random amplification of polymorphic DNA, rep-PCR, arbitrarily primed-PCR, and variable number repeat typing. Rep-PCR amplifies repetitive intergenic sequences, which are then subjected to electrophoresis. Fragment methods use restriction enzymes to digest DNA, and pulsed-field gel electrophoresis is the standard reference-typing method. In sequence amplification, either single-locus or multilocus genes are compared to international standards and both housekeeping and virulence genes can be assessed. In genomic methods, next-generation sequencing is the reference standard because of accurate phenotyping of many genes and advanced computerisation [54]. Because bacteria in biofilms may be in starvation mode and thus less active or are persister bacteria, the best method in the case of biofilms on hospital surfaces and medical equipment will depend on the ability to actually identify biofilms and recover bacteria from them. 

In view of the high cost of biofilms in terms of morbidity, mortality, and medical costs, intensive research on optimal comprehensive and cost-effective methods of detecting biofilms on medical surfaces and equipment is a priority. Even if the detection methods are initially expensive, the costs of biofilms to patients are enormous. 

**Strengths:** The systematic review was conducted with no language or date limits. The study focused on a comprehensive approach to identify both current and novel interventions to detect and destroy biofilms in burn wounds, surfaces, and medical equipment patient contacts in hospitals or LTCFs. 

**Weaknesses:** The key problems are the heterogeneity in the numbers of bacteria tested, comparator interventions, dose ranging studies, replication of experiments, in vivo tests, outcome measures, and microscopy methods. Many studies did not identify if they were testing bacteria that were strong, intermediate, or weak biofilm formers, and thus the effects of interventions could have been incorrectly attributed. Simple metrics such as enough reductions in cfu/mL are not available to rank studies for further large-scale testing. Interventions with positive outcome measures which merit testing are starred *** in Table 1.

## 5. Conclusions 

The essential step is to reduce entry of biofilms into hospitals by establishing isolation rooms and providing the staff to administer tests there to ensure that when patients are admitted they are tested for MDROs, HAIs, and biofilms and are treated before being admitted to the hospital or returned to long-term care homes. The admission of patients with established biofilms needs to be conducted with full precautions. 

An integrated and comprehensive approach is required to detect and eliminate biofilms on both patients and on surfaces and equipment they contact. This review identified 20 interventions that merit further testing and replication in laboratory RCTs and then in large c-RCTs with patients, hospital surfaces, and equipment. The large number of studies required would best be conducted by a consortium of burn centres and infectious disease specialists. These interventions include phages, nano-silver, AgSD-NLs@Cur, silver in the form of Acticoat and Mepilex, acetic acid, graphene-metal combinations, CuCo_2_SO_4_ nanoparticles, Chlorhexidene acetate nanoemulsion, a hydrogel with moxifloxacin, carbomer, Chitosan and Boswellia, light therapy using LEDs with nano-emodin to release reactive oxygen species, blue light + Carvacrol, antimicrobial blue light, mannosidase + trypsin, NCK-10 (a napthalene core compound with a decyl chain appendage), the antimicrobial peptide PV3, which includes two snake venoms, and the polypeptides P03 and PL2. Most of these interventions are aimed at penetrating cell membranes. 

Some current cleaning routines monitor thoroughness of cleaning by environmental service workers by touching random samples of surfaces with invisible markers then use fluorescent light detectors to see if the markers have been removed. The reliability of methods such as blotting and sonification of surfaces to detect biofilms and the effectiveness of their removal needs to be tested. 

The number and types of surfaces and equipment currently cleaned and disinfected needs to be substantially augmented to include keyboards, tablets, cell phones, medical equipment (especially endoscopes), sinks, drains, and food preparation in kitchens.

## Figures and Tables

**Figure 1 ijerph-18-13195-f001:**
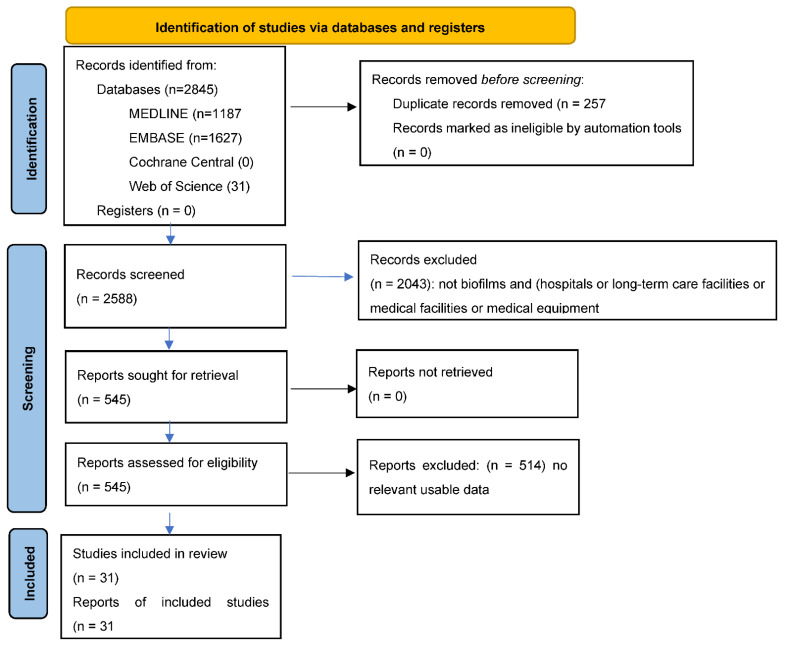
PRISMA Flow diagram for new systematic reviews which include searches of databases and registers only.

**Table 1 ijerph-18-13195-t001:** Reductions in vivo and in vitro biofilms after interventions.

Author, Date	Bacteria and Interventions	Number of Strains Tested and If In Vivo Number of Animals
**Silver compounds**		
Gholamrezazadeh 2018 [6] ***	**In vitro biofilms:** Growth inhibited by Benzalkonium chloride of all *P. aeruginosa* isolates at MBC 0.1 ± 0.2 mg/mL, Deconex MIC 1.0 ± 0.2 mg/L; and nano-silver MBC 28.3 ± 2 mg/mL. **In vitro biofilm formation:** Decreased with bacterial concentration of 10^6^ cfu/mL with nano-Ag 12.5 mg/mL nanomolecule formulation of silver (nano-Ag) at 12.5 ng/mL, causing a reduction in the number of *P. aeruginosa* bacteria forming biofilms from 28.5% to 3.5%, and benzalkonium MIC at 0.03 mg/mL reduced the number of bacteria forming biofilms from 28.5% to 18.7%.	*Pa* 28
Pourhajibagher 2020 [14] ***	**In vitro biofilms:** With MIC_90_ doses of AgSD-NLs@Cur with LED, *A. baumanii* numbers decreased by 76.4%, with AgSD-NLs by 44.8%, and with AgSD 38.1%.	*Ab* 100 *
Halstead 2015 [13] ***	**In vitro biofilms:** Reduction compared to control after 72 h incubation: (a) *A. baumanii 1701* Acticoat 96%*;* Mepilex Ag 95.9%; acetic acid (concentrations 0.31% 5%) 90–93%: *A. baumanii 721* Acticoat 100%; Mepilex Ag 100%; acetic acid 5% (concentrations 0.31% and 5%) 90–93%: (b) *P.* *aeruginosa* 15692 Acticoat 100%*;* Mepilex Ag 100%; acetic acid (concentrations 0.31% to 5%) 86–96%; *P. aeruginosa* 1586 Acticoat 94%*;* Mepilex Ag 99.9%; acetic acid (concentrations 0.31% to 5%) 88–97%.	
**Other metals**		
Karaky 2020 [8] ***	**In vitro biofilms****:** Graphene reduced the biofilm forms of the bacteria significantly more than the planktonic forms (*p* < 0.0001). Eight metal–graphene combinations reduced the amount of intact biofilm by 90% or more (platinum–graphene oxide, gallium–graphene oxide, molybdenum–graphene oxide, gold–graphene oxide, silver–graphene, gallium–graphene and molybdenum–graphene. The greatest reduction in *P. aeruginosa* biofilm metabolic activity occurred with gold–graphene oxide (94%), molybdenum–graphene oxide (93%), silver (91%), and silver–graphene (91%).	*Pa* 2
Li 2020 [15] ***	**In Vitro:** After treatment with H_2_O_2_ (2 nM) and CuCo_2_S_4_ nanoparticles (100 μg/mL) after one hour, 3.6 log reduction was shown in viability of MRSA, 3.3 log in *S. aureus*, and 4.7 log reduction in *E. coli.* H_2_O_2_ and CuCo_2_S_4_ nanoparticles (100 μg/mL) separately showed no biocidal activity. **In vivo** mouse burns: After 2 days treatment with H_2_O_2_ (2 nM) and CuCo_2_S_4_ nanoparticles (100 μg/mL), no inflammatory response and burn wound contracted, and control group showed severe inflammatory response with suppuration. After 6 days group treated with H_2_O_2_ (2 nM) and CuCo_2_S_4_ showed enhanced healing and 83.7% wound closure; CuCo_2_S_4_ 71%, H_2_O_2_ 63.3%, and control 59%. At 2 weeks H_2_O_2_ (2 nM) and CuCo_2_S_4_ wounds completely closed and healed.	MRSA 1 (32 mice)
Nozari 2021 [16]	**In vitro:** After 18 h, ***S. aureus*** with chitosan–alginate–gelatin film for three samples ranged from 1 × 10^4^ cfu/mL to 3.2 × 10^5^ cfu/mL compared to control 1.5 × 10^9^ cfu/mL; with chitosan-bentonite–gelatin film ranged from 7.8 × 10^5^ cfu/mL to 3 × 10^6^ cfu/mL compared to control 1.5 × 10^9^ cfu/mL; (99.99% reduction); (2) ***P. aeruginosa*** with alginate film 1 × 10^4^ cfu/mL to 8.2 × 10^5^ cfu/mL compared to control 1.5 × 10^9^ cfu/mL; with bentonite film 1 × 10^4^ to 1.9 × 10^5^ cfu/mL compared to control 1.5 × 10^9^ cfu/mL; (99.99% reduction) **In vivo mouse burns at 7 days histology:** For the treated rats, there was re-epithelialisation, active fibroblasts, and hair follicles and sebaceous glands were detected. No re-epithelialisation was detected in untreated rats.	*Sa* 1; *Pa* 1; (3 rats)
**Disinfectants**		
Halstead 2015 [17] ***	**In vitro biofilms:** For 23 isolates, acetic acid MBIC 0.31% and MBEC against formed biofilms ranged from ≤0.10% to 2.5% and eradication of mature biofilms was observed for all isolates after three hours of exposure.	*Pa* 9; *Ab* 8; *Sa* 3; *Kp* 2
Song 2016 [18] ***	**In vitro:** Reduced bacterial viability 90% by CNE at 8 μg/mL within 5 min and bacteria completely killed with 8 μg/mL by 1440 min. CHX at 8 μg/mL reduced bacterial viability by 90% at 240 min but bacteria were not completely killed by 1440 min. **In vivo:** Mouse burn wound with CNE 5 mg/mL scab detached from wound on the 8th day and completely detached on the 29th day; CHX 13th and 33rd days. **In vivo biofilms:** As shown by scanning electron microscopy, MRSA biofilms treated with 2 μg/mL CNE biofilms were “dispersed and disrupted and obvious reduction in number of bacteria” and large vacuoles between cell wall and cytoplasm. Dead/live cell ratio with CNE 83.6%, with CHX 13%.	MRSA 1 (mice; n = ?)
Tiwari 2018 [19]	**In vitro biofilm** % **reduction in biofilm optical density (OD):** No significant differences between reductions in strong and weak biofilm formers for either sodium hypochlorite or ethanol. With 0.6% sodium hypochlorite for strong biofilms, 34.27% ± 15.30, and for weak biofilms, 35.07 ± 12.98 (*p* = 0.897); (2) with 70% ethanol for strong biofilms, 18.14% ± 11.56 and for weak biofilms, 20% (*p* = 0.488). On electron microscopy, strong biofilm producers showed significant depressions and irregular craters on their surface.	*Sa* 29 *
Andersson 2021 [9]	***S. aureus*** (1) **Wound surface reductions***S. aureus* from cfu 10^8^ to cfu log 10^6^ for levofloxacin 2 μg/mL compared to control cfu increased to log 10^10^ (*p* < 0.0001); (2) **wound tissue reductions** to cfu log 10^6^ for levofloxacin 2 μg/mL compared to control cfu Log 10^8.5^ (*p* < 0.001); (3) **wound surface reductions** to cfu log 10^6^ for Prontosan compared to control cfu log 10^10^ (*p* < 0.0001); (4) **wound tissue reductions** to cfu log 10^7^ for Prontosan compared to control cfu log 10^8^ (*p* < 0.05) ***P. aeruginosa*** (1) ***P. aeruginosa* wound surface** no change in cfu log 10^8^ for levofloxacin 2 μg/mL, but control increased to cfu log 10^11^ (*p* < 0.001); (2) **wound tissue** no change cfu log 10^8^ for levofloxacin 2 μg/mL and control cfu log 10^8^ (n.s.); (3) **wound surface** reduction from cfu 10^8^ to cfu log 10^6^ for Prontosan compared to increase in control to cfu log 10^11^ (*p* < 0.001); (4) **wound tissue** reductions to cfu log 10^7^ for Prontosan compared to control cfu log 10^8.5^ (*p* < 0.05)	*Sa* 1; *Pa* 1 (pigs n = ?)
Chhibber 2020 [20] ***	**In vivo biofilm:** (1) Conventional hydrogel 2.8 log_10_ cfu/mL reduction on day 1; 4.2 log_10_ cfu/mL reduction day 2; and wound became sterile (day not stated). (2) Novel hydrogel 3.5 log_10_ cfu/mL reduction on day 1, 4.8 log_10_ cfu/mL reduction day 2, and wound sterile (day not stated). (3) Control 6.9 log_10_ cfu/mL count day 3. (4) At 4 h, complete eradication of MRSA from wounds with conventional and novel hydrogels but MRSA established in control mice.	MRSA *b* 1 (54 mice)
**Sonotherapy**		
Pourhajibagher 2021 [21] ***	**In vitro biofilms:** Reduction in multi-species bacterial growth following SDT at ½ MIC of N-EMO was 81.5 %; at 1/16 MBIC 71.0%; and at 1/128 MBEC 57.8; (reductions in log_10_ cfu/mL 99.99%, 99.97%, 99.48%)	*Pa* 1; *Sa* 1. *Ab* 1
**Light therapy**		
Ishiwata 2021 [22]	**In vivo: Baseline:** 8.9 × 10^4^ cfu/mL; **Day 0 post infection**: aPDT group, no bacteria, control 3.4 × 10^8^ cfu/mL; **Day 1**: aPDT 3.5 × 10^5^ cfu/mL, PS 4.7 × 10^5^ cfu/mL indicating rapid regrowth; **Days 2–7**: rapid regrowth each day and aPDT group day 6 × 10^4^ cfu/mL **Rat survival at day 7:** aPDT 11/14. PS 3/10, control 2/14	*Pa* 1; (34 rats)
Lu 2021 [3] ***	**In vitro biofilms:** *A.b*, *P.a**AF0001 and* MRSA IQ0064 biofilms at 10^7^ cfu/mL completely eliminated after 22.5 min of blue light + carvacrol (*p* < 0.0001) and reduced Ab biofilm from 58.6 μm to 1.4 μm thickness and MRSA from 32.4 μm to 1.7 μm; six first-line antibiotics inactivated < 1.5 log CFU after 6 h. **In vivo mouse burns:** Carvacrol 50 μL at 1 mg/mL + blue light for 12 min (40 J/cm^2^) with luminescent bacteria eliminated log 8 luminescence, blue light alone 2.3 log, and carvacrol 0.8 log.	*Ab* 1 *Pa* 1 MRSA 1; (mice n = ?)
Pourhajibagher 2020 [2]	**In vitro:** Reduction in cell viability by ICG at 1000 μg/mL significant reduction in cell viability of *A. baumannii* 1.5 × 10^5^ cfu/mL, *P. aeruginosa* 1 × 10^5^ cfu/mL; *S. aureus* 1.0 × 10^5^ cfu/mL compared to control 4.5 × 10^5^ cfu/mL (all *p* < 0.05).	*Ab* 1, *Pa* 1, *Sa* 1
Wang 2016 [4] ***	**In vitro:** (1) Exposure of 24 h old and 72 h old *A. baumanii* biofilms to aBL 432 J/cm^2^ for 72 min resulted in inactivation of 3.59 log_10_ and 3.18 log_10_ cfu/mL. (2) Exposure of *P. aeruginosa* biofilms to aBL 432 J/cm^2^ resulted in inactivation of 3.02 log_10_ cfu/mL and 3.12 log_10_ cfu/mL. Control biofilms showed <0.27 log_10_ cfu/mL loss of viability for *A. baumanii* and <0.42 log^10^ cfu/mL for *P. aeruginosa.* **In vivo:** Mouse burn wounds infected with 5 × 10^6^ cfu/mL *A. baumanii* at 24 h required 360 J/cm^2^ at 48 h 540 J/cm^2^ to inactivate 3 log_10_ cfu/mL in biofilms.	*Ab* 1, *Pa* 1; (mice n = ?)
**Small molecules**		
Banar 2016 [23] ***	**Minimum biofilm eradicating concentration (MBEC):** Strain 1: ceftazidime (CAZ) 1024 μg/mL, CAZ + α-mannosidase 128 μg/mL, CAZ + β-mannosidase 128 μg/mL, CAZ + trypsin 512 μg/mL; Strain 2: ceftazidime (CAZ) 1024 μg/mL, CAZ + α-mannosidase 4 μg/mL, CAZ + β-mannosidase 4 μg/mL, CAZ + trypsin 8 μg/mL; Strain 3: ceftazidime (CAZ) 1024 μg/mL, CAZ + α-mannosidase 4 μg/mL, CAZ + β-mannosidase 8 μg/mL, CAZ + trypsin 32 μg/mL. All tested concentrations killed biofilm bacterial cells.	*Pa* 57 *
Ghosh 2015 [24] ***	**Persister cells:** NCK-10 completely lysed persister cells of 5 log cfu/mL *E. coli* after 2 h, but colonies persisted in control group at 5 log cfu/mL. **Disruption of biofilms:** EC_50_ = 30 μM against biofilms of *A. baumanii* (MTCC 1425); 20 μM against *E. coli MTCC 443)*; 26 μM against *K. pneumoniae (ATCC 700603)*, and 19 μM against *P. aeruginosa (MTCC 424).* On confocal microscopy in the treated samples, the biofilms were completely disrupted, and the untreated samples had biofilms 12.6 μm thick.	*Ab* 3; *Pa* 3; *Ec* 3; *Kp* 2; (20 mice)
Goodwine 2019 [1] ***	**In vitro** (1) Samples from human wounds: 2.2-fold reduction after exposure to 5 mU pyruvate-dehydrogenase (PDH) and by 2.9-fold after 10–20 mU; **In vitro biofilms** (1) On confocal laser scanning microscopy 60% of microcolonies in PDH-treated biofilms showed signs of dispersion with central voids and 8% of untreated biofilms. (2) Four-day old human wound samples of *S. aureus* biofilms exposed to PDH 10 mU had 40% reduction in mass. **In vivo** (1) Pig burn wounds: *P. aeruginosa* biofilm population mass reduced 2-log with tobramycin 200 μg/mL compared to untreated control. (2) A 4-log reduction by tobramycin 200 μg/mL + PDH 200 mU compared to control. (3) Silver sulfadiazine 2-log reduction in biofilm and 4-log in planktonic populations.	*P.a* 1; (3 pigs)
Han 2018 [25] ***	**Biofilms: reduction in** cell viability of *S. aureus* CICC10790 to 10% with 8 × MIC vancomycin (8 μg/mL) and to 10% with AMP-jsa9 at 8 × MIC (128 μg/mL); reduction in biomass to 15% with 8 × MIC vancomycin (8 μg/mL) and to 15% with AMP-jsa9 at 8 × MIC (128 μg/mL). **In vivo:** In mouse scalded skin burns viable cell count treated with vancomycin or AMP-jsa9 were 10^1^ to 10^2^ on days 3 and 7 and in those treated with kanamycin or saline 2–3 × 10^4^ at 3 days and 5–6 × 10^5^ at 7 days with a large infiltrate of inflammatory cells.	MRSA 1; (mice n = ?)
Konai 2020 [5] ***	**In vitro biofilms:** With confocal scanning electron microscopy, D-LANA-14 8 μg/mL plus 8 μg/mL colistin resulted in >80% reduction in biofilm mass of *A. baumanii*-R674 and *P. aeruginosa*-R590; D-LANA-14 8 μg/mL showed no effect, and rifampicin 8μg/mL showed 25–30% disruption. **In vivo:** Burn wounds in mice: D-LANA-14 40 mg/kg plus rifampicin 40 mg/kg caused 4.9 log reduction in *A. baumanii*-R674 and 4.0 log in *P. aeruginosa*-R5902; D-LANA-14 2.3 log and 1.3 log; and rifampicin 3.0 log and 1.6 log.	*Ab* 3; *Pa* 5; (mice n = ?)
Memariani 2016 [26] ***	**In vitro biofilms:** Scanning electron microscopy with acridine-orange/ethidium bromide staining: PV3 treated cells were shorter, blisters on membranes, roughness, and blebbing. For PV3 at 8 × MIC at 24 h resulted in “almost” 100% killing of cells and 95% biomass removal.	*Pa* 7
Pan 2020 [27] ***	**In vitro biofilms:** P03 reduced the biomass of *P. aeruginosa* biofilms by 76.9%, PL2 by 35.1%, and PH2 by 31.45%, Polymixin by 7.8%. **In Vivo**: In mice burn wounds: P03 caused 78.2% reduction in *P. aeruginosa* and PL2 caused 49.3% reduction compared to Polymixin B.	*Pa* 1; MRSA 1 (mice n = ?)
Su 2019 [28]	**In vitro biofilms:** Microtitre dish biofilm formation assays: after 2 μg/mL PTM or PTM-2t biofilm formation for *S. aureus* ATCC 291213 reduced 95%. **In vivo:** Mouse burns treated with 4 mg of PTM or PTM-**2t** on burn wound twice daily × 7 days. PTM reduced *S. aureus* to 2 × 10^6^ cfu/g and PTM-**2t** to 8.6 × 10^6^ cfu/g compared to 2.5 × 10^6^ cfu/g for mupirocin and untreated mice 4.3 × 10^8^ cfu/g.	MRSA 1, *Sa* 1; (20 mice)
Uusitalo 2017 [29]	**In vitro biofilms:** (1) INP0321 at 100 μM reduced biofilm to 40% of control (*p* < 0.05); (2) INP0341 inhibited *P.* *aeruginosa* swarming and prevented movement across semisolid surfaces which requires flagella and type IV pili. **In vivo:** Treated mice died at 36 h, controls as 42 h (*p* < 0.05)	*Pa* 1; (mice n = ?)
**Glycans**		
Wheeler 2019 [30]	**In vitro:** (1) *P. aeruginosa* PA01 biofilms exposed to mucins 70% of cells dissociated from surface into planktonic phase (*p* < 0.0001). (2) MUC5AC and MUC5B 0.5% *w*/*v* suppressed virulence pathways 1, 2, 3, and 6 secretion systems; siderophore biosynthesis; pyoverdine and pyochelin; and quorum sensing. (3) MUC5AC suppressed *P. aeruginosa* PA01 association with plastic and glass surfaces and attachment to live HT human epithelial cells in a concentration dependent manner. **In vivo****:** Pig burn wounds with MUC5AC 1 week post infection, two-log reductions in *P. aeruginosa* CFUs, no reduction without mucins.	*Pa* 1; (4 pigs)
**Lactobacilli**		
Lenzmeier 2019 [31]	**In vivo:** LgCS inhibited the growth of P. aeruginosa strain PAO1, reduced biofilm development 40-fold at 8 h (control significantly increased), and eliminated biofilms at 28 h. **In vitro:** Mouse burns: local treatment of wound by LgCS did not inhibit *P. aeruginosa* growth in wound at 24 h but prevented transfer to blood stream with 100% survival of mice at 7 days treated with LgCS (no *P. aeruginosa* in livers or spleens), 100% death due to sepsis in untreated mice (~10^7^ cfu/mL *P. aeruginosa* g^−1^ in livers and spleens). Second dose of LgCS 24 h after first dose completely eliminated *P. aeruginosa* in wound.	*Pa* 1; (20 mice)
**Phage Therapy**		
Alves 2018 [33]	**Ex vivo biofilms:** 24 h after phage treatment, phage treated 10^6.5^ cfu/mL compared to control 10^7.5^ cfu/mL, (*p* ≤ 0.0001); 48 h after phage treatment 10^7^ cfu/mL compared to control 10^7^ cfu/mL (n.s.);	MRSA 27; (pig skins, not live pigs, n = ?)
Ho 2016 [34] ***	**In vivo:** Carbapenem-resistant *Acinetobacter baumanii* (CRAB) 8.57/1000 patient days pre-intervention, 5.11 during aerosol phage intervention period (*p* =.0029), resistant isolates decreased 87.6% to 46.07% (*p* = 0.001) **Decreased drug use:** colistin 7876 DDD/1000 patient days decreased to 3158 (*p* =0.0177); tigecycline 2737 to 753 (*p* = 0.0005); meropenem 5084 to 2469 (*p* = 0.0385); imipenem 1384 to 1101 (ns).	
Holguín 2015 [35]	**In vitro:** at 18 h after phage therapy, P1 10^7.5^ decreased to 10^4^ pfu/mL, P2 10^8^ to 10^4.5^, P4 10^7.5^ to 10^2.5^ (by visual inspection of Figure 2 in Holguín’s article), P2 not reported) **In vitro biofilms:** P1 17% reduction at 0 h (*p* = 0.003), 34% at 24 h (*p* = 0.134), 55% at 48 h (*p* = 0.005), P3 59% reduction at 0 h (*p* = 0.00001), 56% at 24 h (*p* = 0.034), 75% at 48 h (*p* = 0.0004), P4 68% reduction at 0 h (*p* = 0.015), 15% at 24 h (*p* = 0.036; 21% at 48 h (*p* = 0.286) **In vivo****:** Φ*Pan70* immediately after *P. aeruginosa* infection 4/5 mice survived; Φ*Pan70* 45 min after infection 5/5 survived; 24 and 48 h after infection 4/5 mice survived; controls all mice died days 3 or 4.	
O’Flaherty 2005 [36] ***	**In vitro:** 14/28 *S. aureus* strains sensitive to phage K 10^7^ cfu/mL and no bacteria remained after 2 h; no bacteriophage-insensitive mutants (BIMs) after 25 h **In vivo:** (1) MRSA strain DPC5645 decreased within 2 h from 5.7 × 10^6^ cfu/mL to undetectable levels; (2) MRSA strain DPC5246 on skin reduced 100-fold with phage K 1.4 × 10^8^ pfu/mL (10 replications, no statement of numbers of participants or hands)	
Pallavali 2021 [37]	**In vitro biofilms****:** At 96 hours after 4 h phage therapy optical density (OD, which corresponds to biomass): (1) *P. aeruginosa* 0.47 ± 0.035 decreased to 0.17 ± 0.024; (2) *E. coli* 0.47 ± 0.035 decreased to 0.15 ± 0.026; (3) *K. pneumoniae* 0.47 ± 0.035 decreased to 0.17 ± 0.022; (4) *S. aureus* 0.47 ± 0.036 decreased to 0.16 ± 0.032. **In vitro confocal microscopy:** Predominant numbers of dead cells after 4 h phage therapy	

* described as “clinical isolates” and not strains. Ab = Acinetobacter baumanii; Ec = E. Coli; Kp = Klebsiella pneumoniae; Pa = Pseudomonas aeruginosa; Sa = Staphylococcus aureus. *** merits further replication in animal and human studies.

## Data Availability

Not applicable.

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
