# Peer review of "Reducing Biofilm Infections in Burn Patients’ Wounds and Biofilms on Surfaces in Hospitals, Medical Facilities and Medical Equipment to Improve Burn Care: A Systematic Review"

_ijerph, 2021, doi:10.3390/ijerph182413195_

Round 1
Reviewer 1 Report
The paper is very interesting. However small changes should be made:
- Some sentences are too long. Please correct it.
- There are quite a few language errors, please read the manuscript carefully.
- The introduction should be shortened, some information can be moved to other subchapters.
- What species of bacteria are prevalent on surfaces in hospitals, medical facilities and medical equipment? You might want to put up a table with this.
- Please consider adding a subsection on methods of bacterial identification (e.g. classical culture, capillary electrophoresis - work by Klodzinska et al., etc.).
Reviewer 2 Report
I have read with interest the Systematic Review by Thomas and Thomas on the different interventions to reduce the biofilms infection in burn patients' wound and medical facilities/equipments.
I have appreciated very much all the efforts made by the Authors, since it is really a substantial paper, that deeply analyze many different interventions (i.e. small molecules, lights and so on).
Neverthless, in my opinion, the paper is too much long and rich, with the risk to become a bit "heavy" for the readers. I am concerning the Tables that would need to be resumed (in its present form, they consist of 20 pages in total!!). The Table should represent a sort of summary of the text, where the reader could easily find a quick look (and resume) of the corresponding text.
So, my first suggestion is to strongly shorten the Tables.
Afterward, I have some minor comments, that are listed below:
- Abstract: some commercial devices are cited (i.e. Acticoat, Mepilx), and the same in the Results sections. It is possible to cite them as for their main components (at least in the Abstract)? Anyway, the Authors need to add the Financial Disclosure at the end of the text.
- Materials & Methods: please clarify why Web of Science was not included in the Databases, since it is of good importance in Medical Sciences. Another one is the Cochrane Collaboration, but maybe this is not so appropriate in a systematic review.
- Figure 1: PRISMA Flow diagram: please pay attention to the spelling/punctuation: number of Medline records identified (10221187?)? Please check the brackets at the end of the lines.
- Results: line 189: in vivo biofilm studies in humans, please add what kind of studies were considered (original articles, case reports...were the reviews included?)
- Results: in some reported studies the bacterial biofilms after interventions were detected by microbiological assays (i.e. cell viability), while in other through electron microscopy or confocal microscopy. Please include what are the possible differences (if any) in the results and their interpretations, with such diverse methodologies.
- Table 1: please respect the same order of the text: the study by Ho 2016 on Phages in human patients should be reported together with Phage therapy. Furthermore, please insert the different considered agent (i.e. mannosidase, D-LANA 14) as a short subgroup in the Small Molecules section of the table
- Conclusions: is it possible to conclude which interventions could be preferred against the main different bacterial biofilms (for instance against S. aureus, A. baumanii, P. aeruginosa and so on) on the basis of the reviewed articles? Please clarify, thank you
Round 2
Reviewer 2 Report
The Authors have addressed all the questions.
I only suggest to pay attention to the "non-published materials" since they have not been updated ,(i.e. the WoS and Cochrane searchs are not reported).
Finally: Table 2 is now in the Appendix or in the Supplementary Materials? (I personally suggest the second one)
Author Response
Thank you for reading and commenting on the revision. As you advised searches were made in WoS and Cochrane Central but neither turned up new trials for inclusion. The WoS as usual had two interesting articles, one on advances in optical detection of biofilms, and one on how to conduct a molecular investigation of hospital bacterial outbreaks, both of which we included.
We agree, the detailed table is too long as an Appendix and on your suggestion has been changed to Supplemental Table 1.